# Biosecurity and Hygiene Procedures in Pig Farms: Effects of a Tailor-Made Approach as Monitored by Environmental Samples

**DOI:** 10.3390/ani13071262

**Published:** 2023-04-05

**Authors:** Annalisa Scollo, Alice Perrucci, Maria Cristina Stella, Paolo Ferrari, Patrizia Robino, Patrizia Nebbia

**Affiliations:** 1Department of Veterinary Sciences, University of Torino, 10095 Grugliasco, Italy; 2CRPA Research Centre for Animal Production, 42121 Reggio Emilia, Italy

**Keywords:** biosecurity, pig, tailor-made plan, ATP rapid test, hygiene procedures, livestock-associated methicillin-resistant *Staphylococcus aureus*, extended-spectrum β-lactamase producing *Escherichia coli*

## Abstract

**Simple Summary:**

This study describes the improvement of biosecurity and environmental hygiene procedures in 20 pig farms monitored during a 12-month period. A checklist was used to develop tailor-made plans, which also included personnel training on hygiene procedures. Adenosine triphosphate (ATP), the content in environmental samples, was used as an output biomarker. To gain an insight into the environmental samples, the presence of livestock-associated methicillin-resistant *Staphylococcus aureus* (LA-MRSA) and extended-spectrum β-lactamase producing *Escherichia coli* (ESBL-*E. coli*) was also investigated as sentinel microorganisms to monitor antibiotic resistance. After 12 months, the average biosecurity was improved, and ATP contents decreased. Despite this, only ESBL-*E. coli* prevalence was effectively decreased by hygiene procedures, and a challenging persistence of a high prevalence of LA-MRSA after cleaning emerged. Results suggest that a tailored approach and on-farm training are useful to improve the application of biosecurity measures, in particular those related to hygiene management in the professional zone. However, this should not reduce the attention to the presence of resistant bacteria in the pig barns, in particular for the risk of spreading these bacteria to humans in close contact with pigs, moving the attention to the healthcare of workers in a one-health approach.

**Abstract:**

In livestock, the importance of hygiene management is gaining importance within the context of biosecurity. The aim of this study was to monitor the implementation of biosecurity and hygiene procedures in 20 swine herds over a 12-month period, as driven by tailor-made plans, including training on-farm. The measure of adenosine triphosphate (ATP) environmental contents was used as an output biomarker. The presence of livestock-associated methicillin-resistant *Staphylococcus aureus* (LA-MRSA) and extended-spectrum β-lactamase producing *Escherichia coli* (ESBL-*E. coli*) was also investigated as sentinels of antibiotic resistance. A significant biosecurity improvement (*p* = 0.006) and a reduction in the ATP content in the sanitised environment (*p* = 0.039) were observed. A cluster including 6/20 farms greatly improved both biosecurity and ATP contents, while the remaining 14/20 farms ameliorated them only slightly. Even if the ESBL-*E. coli* prevalence (30.0%) after the hygiene procedures significantly decreased, the prevalence of LA-MRSA (22.5%) was unaffected. Despite the promising results supporting the adoption of tailor-made biosecurity plans and the measure of environmental ATP as an output biomarker, the high LA-MRSA prevalence still detected at the end of the study underlines the importance of improving even more biosecurity and farm hygiene in a one-health approach aimed to preserve also the pig workers health.

## 1. Introduction

The concepts of disinfection and what has come to be known as biosecurity are not new. Since the beginning of written history, humankind has recorded guidelines to deal with disinfection and biosecurity based on a time-specific understanding of the concepts. Throughout history, the concepts of “clean” and “unclean” have become increasingly refined. Today, a disinfectant, also synonymously and generically referred to as a germicide, is defined as “an agent that destroys infection-producing organisms” [1]. A single product is unlikely to satisfy all disinfection requirements in a given facility, and the realistic use of disinfectants will rely on multiple factors, including the degree of microbial killing required, the nature and composition of the surface, item or device to be treated, the cost, safety, and ease of use of the available agents [2]. However, it should be pointed out that a considerable reduction of infectivity may be achieved by simple cleaning, which represents one of the most important steps in the entire process, whose goal is the removal of more than 90% of microorganisms and when it is properly performed, it greatly improves the disinfection efficacy [3].

In swine farm premises, the importance of hygiene procedures as a biosecurity vehicle is recently increasing. For example, after the African swine fever virus became the putative most relevant epizootic disease considering an animal health perspective, the recent European regulation laying down special control measures for this disease highlighted the importance of good cleaning and disinfecting procedures in farm facilities and in the vehicles used for the transportation of pigs [4]. These recommendations follow the principles already published in the older Council Directive 2002/60/EC [5]. Biosecurity is defined as the combination of all the measures and procedures whose application on-farm lead to a reduced risk of introduction and spread of an infectious agent [6]. Therefore, a detailed assessment of risk factors that characterize a farm is necessary to obtain a successful identification of biosecurity measures to be implemented, including hygiene procedures.

Unfortunately, motivating farmers to change their daily routines and turn to a highly accurate hygiene protocol is a well-known challenge [7,8], and cleaning procedures, as well as biosecurity measures in general, seem particularly difficult to implement only by legal requirement [9]. The importance of knowledge in influencing behaviour is widely recognised, as individuals need to be aware of the consequences of their actions before they can adjust their attitudes towards a particular challenge [10]. It has been suggested by several authors that failure to comply with biosecurity measures is often attributed to inadequate training of farm personnel and poor communication between farm workers and technical services [11]. This is especially true in terms of understanding the significance of each measure in relation to disease transmission. To bridge this gap and enhance communication between farmers and biosecurity advisors, Scollo et al. [12] proposed the implementation of a customised biosecurity plan.

The objective of the study was to obtain a better understanding of the implementation and enhancement of biosecurity in a convenience sample of 20 pig farms through the development of tailor-made herd protocols. Continuous and individual training of the farms’ personnel on cleaning and disinfection procedures was included. Improvement of biosecurity was monitored over a 12-month period, and the farm’s hygiene status was evaluated using adenosine triphosphate (ATP) environmental content as a biomarker and output parameter. A second goal was to investigate and describe the presence of livestock-associated methicillin-resistant *Staphylococcus aureus* (LA-MRSA) and extended-spectrum β-lactamase (ESBL) producing *Escherichia coli* in the environment before and after hygiene procedures throughout the study. These microorganisms have been chosen because they are widespread among different kinds of farm animals as well as in humans that have close contact with animals [13,14,15] or live in regions with a high pig density [16] and are considered sentinel microorganisms to monitor antibiotic resistance [17]. The presence of other methicillin-resistant members of the Staphylococci group (MRS) was also investigated, as they may have a role as a reservoir of antibiotic-resistance genes in some human infections.

In this way, the study also improves the limited studies regarding the effect of a biosecurity and hygiene tailor-made plan on the detection of these isolates.

## 2. Materials and Methods

The present study was conducted as a part of a larger research program called “Healthy livestock, tackling antimicrobial resistance through improved livestock health and welfare”, a European Union’s Horizon 2020 research and innovation program. The study proposes a method, previously implemented in pig farms by Scollo et al. [12], that considers both input and output parameters to assess the exposure and the risks related to the introduction and spread of infectious diseases in commercial pig farms. Input parameters were information on the biosecurity status of the farm, collected by a Biosecurity Risk Analysis Tool (BEAT), which assessed the likelihood of pathogen introduction, susceptibility of animals to exposure, and disease transmission within the farm. As an output parameter (i.e., an indicator obtained from the observation of the animals or the environment, which is monitored during the time for the early detection of negative events after biosecurity breaches), the ATP environmental content was selected.

For the selection of the farms to be involved in the study, recommendations of de Oliveira Sidinei et al. [18] were adopted in order to minimise possible influences of contractual conditions on the biosecurity status or aptitude at the beginning and during the study. In total, 20 farms were randomly selected from a list of farms with the same agreement signed with the same contractor located in Northern Italy (15,000 km^2^ distributed in the regions of Piedmont, Lombardy, and Emilia Romagna). This area, characterised by an average temperature during the solar year ranges from 12 to 14 °C and precipitation up to 1400 mm [19], was selected for the high concentration of pigs designed to the most typical Italian pork product, the Protected Designation of Origin (PDO) ham. The area accounts for 80% of the national pig production [20]. For each farm, further information about the production phase (breeding/nursery: from the farrowing site to approximately 30 kg of body weight; fattening: from approximately 30 kg of body weight to slaughter-170 kg of body weight) and herd size (in case of breeding/nursery: number of productive sows and/or number of weaned piglets present the day of the visit; in case of fattening sites: number of farmed pigs present the day of the visit) were collected.

### 2.1. Development of Tailor-Made Programs

#### 2.1.1. The BEAT Questionnaire

Before being recruited, every farm was contacted by phone and informed about the project. All the farms joined the project, and over the course of a 12-month study period, the farms were visited three times. The biosecurity asset of the 20 pig farms was described using the BEAT questionnaire as part of the project described above. Details of the checklist are available in Scollo et al. [12] (see also the checklist template in the Appendix A). The questionnaire, taking into account the FAO three-zone biosecurity model [21], covered several relevant aspects of biosecurity with the aim of ascertaining the implementation of preventive measures and the presence or absence of specific circumstances. The BEAT included five main sections related to both external (risk of farm colonisation by new pathogens) and internal (pathogen spread among different barns and/or areas of the farm) biosecurity: the public zone (outside the farm perimeter), the professional zone (between the animal barns), the herd zone (inside the animal barns), and the two interfaces between public/professional zones and professional/herd zones. There were in total 97 items in the five biosecurity sections of the questionnaire, each one assessed on a 4-point scale: from 0 to totally inadequate items, to 3 to completely satisfying items. This rating was divided into 1-point linear increments [22,23,24]. The historical and actual biosecurity status was collected through the BEAT application on the first visit to each farm, with the aim of identifying critical items in the biosecurity layout. Thereafter, information was used to develop a tailor-made plan to drive the farm towards the implementation of biosecurity. The development of the plan was set up by the researcher leading the visit together with the farm manager, discussing the details of the plan also with the farm’s personnel, following suggestions from Donaldson [25]. The customisation of the plans included farm-specific recommendations adapted to the farm context and more likely to meet farmers’ objectives [26,27,28]. Among the specific characteristics of the farm context, the socio-economic situation was particularly considered. This was aimed to make farmers more likely to perceive the benefits and the feasibility of these recommendations, ensuring that the biosecurity plan can realistically be implemented. After 6 months, ongoing monitoring of compliance with biosecurity plans was verified through a second visit. If necessary, the written plan was modified or updated [25]. The biosecurity status was re-evaluated using the BEAT application after a period of 12 months from the initial visit (Figure 1). As recommended by Dewulf and Immerseel [29], farmers were interviewed in person before inspecting the farm. The interview lasted 1.5 h on average. Thereafter, the inspection of the farm was carried out to enable a comparison between the farmer’s response and the current situation on-site. It took an average of one hour to visit the farm. If the answers to the questionnaire were not accurate, the farmer was notified, and the given answer was modified. Interviews and farm visits were conducted by the same qualified veterinarian with professional experience in swine production to minimise interviewer bias as much as possible and to ensure inter-farm comparability. A maximum of two farms were interviewed per day.

#### 2.1.2. Hygiene Management: Training and Raising Awareness

At the beginning of the study, all farm managers and employees involved in cleaning and disinfection procedures were invited for hygiene management training. At least one employee involved in cleaning and disinfection procedures from each farm participated, with a total number of 29 attendees. The training started with an introductory lecture regarding basic protocols of hygiene management, the differences in cleaning and disinfection, and biological foundations of microbiological contaminants in a 60 min oral presentation with PowerPoint presentation support. The lecturer was an expert technician in hygiene management, and the same lecture was repeated five times in different geographical areas (provinces of Modena, Brescia, and Mantova) to allow attendees to join the most suitable scheduled meeting, depending on their personal/working calendar. The introductory lectures were held in five different meeting rooms belonging to five farm properties selected among the 20 involved farms; mandatory characteristics for the selection of the location were the meeting room availability, proximity with a pig barn, presence of a dressing room for visitors, and availability of the owner to host the event. The number of attendees per lecture ranged from 5 to 7. Right after the lecture, attendees were invited to participate in a practical cleaning and disinfecting session on-farm in the adjacent barn under the guidance of the expert technician. Before entering the barn, good biosecurity practices for visitors were applied. During this session, a practical demonstration of correct procedures and common errors/mistakes was organised, and all the participants were contextually involved in a chaired group discussion and encouraged to compare their experiences suggesting measures to improve the hygiene status. The correct procedures that were illustrated were based on the protocol described by De Lorenzi et al. [30] for commercial pig holdings. Special emphasis was dedicated to listing and investigating all the farm areas that need a good cleaning and disinfection protocol: barns, hygiene lock, loading bay, carcasses storage, and equipment. Briefly, the protocol started with the cleaning phases: dry cleaning (the removal of all the residual organic material—e.g., food, faeces, litter, dust—from the barn and its equipment), pre-soaking (by pressure washers with a high-water flow but low pressures), and wet cleaning (washing with warm water and detergent by foaming, and leave the detergent for the labelling recommended period). After the cleaning phase, all surfaces were rinsed with cold water in order to remove detergents and all traces of material used in the cleaning process. Thereafter, surfaces were dried completely before the application of the disinfectant. Surface disinfection was carried out with high pressure by heating a highly concentrated disinfectant, subsequently converted to fog by a fogger (thermal fogging). Suggested contact time was 24 h, with the recommendation to apply the disinfectant again to keep the surface wet for the required contact time in case of faster drying (e.g., due to weather). Premises should remain empty for 7 days after drying the disinfectant in order to avoid the accidental absorption of residues by the animals. No specific detergent or disinfectant was suggested, but compliance with instructions on the label was always recommended (e.g., concentration in the water solution).

Additionally, during the 12 months of the study, each farm was visited by the expert hygiene technician at least one time during the cleaning and disinfection process, with the aim of discussing the weaknesses or strengths of the procedures.

### 2.2. Environmental Samples

Environmental measurement of pathogen presence was investigated through swab samples during the first and the third visit to verify hygiene levels. Procedures described by Heinemann et al. [31] were adopted.

Five sampling sites were tested on each farm from a single and randomly selected pen (or farrowing pen, in the case of breeding farms) that was representative of the farm. These sampling sites included the corner of the dunging area, the upside and inside of the feeding tube, one nipple drinker, the outside and inside of the trough, and one manipulable material (environmental enrichment). The pen was selected as the best area to test hygiene procedures as the dirtiest one, considering the animals’ contribution to organic material during their life cycle (faeces, oral fluids, etc.). To collect samples from planar surfaces, the area was wiped horizontally and vertically, covering 25 cm^2^ at each sampling site. The inner nipple and the outer tube of drinkers and feeders were swabbed in a circular motion. Swabs were pre-moistened with sterile physiological saline (0.5 mL/sample).

#### 2.2.1. ATP Rapid Test

Environmental measurement of the ATP content was investigated in all the farms (*n* = 20). One sample for each sampling site was immediately tested on the farm for the analysis of ATP content (CleanTrace Surface ATP Test Swab UXL100, 3 M, Neuss, Germany). The samples were taken 7 days after cleaning and disinfection procedures (after drying of the disinfectant), before restocking, for a total of 5 samples per farm. This test system relies on a bioluminescence response that uses ATP as a cofactor. After gently scrubbing the targeted area, the ATP test was activated by pushing down the stick shaft to remove the membrane and starting the enzymatic reaction. After 10 s of shaking, ATP bioluminescence was measured in relative light units (RLU) by a luminometer (NG III, 3 M). The resulting values were displayed in log10 RLU/cm^2^. Samples collection was made during the first and the third (last) visit for the BEAT application in the same sampling pen (Figure 1).

#### 2.2.2. Bacteriological Investigation

In a subsample of 10 out of the 20 farms involved in the study, two samples for each sampling site (except the feeding tube) were collected during the first visit and addressed for bacteriological investigation: one for MRS and one for ESBL-*E. coli*. The subsample of farms was sorted by a convenience selection to maintain the same geographical distribution of the original sample of 20 and to involve both the production phases (breeding/nursery and fattening). A general description of the subsample of farms is provided in the Appendix A. The samples were taken before cleaning and disinfection procedures (in a dirty environment, right after emptying the pen from animals) for a total of 8 samples per farm. Seven days after cleaning and disinfection procedures (after drying of the disinfectant), before restocking, a second identical sample collection (*n* = 8 per farm) was repeated in the same pen. Each sample was identified with a code indicating the progressive farm identification number (1, 2… 10), sampling site (1 = nipple drinker; 2 = trough; 3 = corner in the dunging area; 4 = manipulable material), bacterial genus to be investigated (S = *Staphylococcus*; E = *Escherichia coli*), and cleaning stage (1 = before hygiene procedures, right after emptying the pen from animals; 2 = after complete hygiene procedures, still on the empty pen). Samples were stored in a refrigerated box until delivery to the laboratory, where they were handled within 24 h from the collection. Six farms adhered to the repetition of sample collection and analysis during the third (last) visit in the same sampling pen (Figure 1).

##### Methicillin-Resistant Staphylococci (MRS) Collection

The isolation of MRS was performed as described by Bonvegna et al. [32] with slight changes. A liquid medium was used to perform an enrichment step on each swab. The medium consisted of Tryptic Soy Broth (TSB) supplemented with 2.5% NaCl, cefoxitin (3.5 mg/L), and aztreonam (20 mg/L) from Sigma-Aldrich, St. Louis, MO, USA. Cefoxitin was added to select MRS, while aztreonam and low salt were included to inhibit the growth of Gram-negative bacteria. The swabs were soaked for three minutes in 5 mL of the broth, and the samples were shaken at 240 rpm for 24 h at 37 ± 1 °C in a shaker incubator. After this enrichment process, a 10 µL loop of the liquid samples was spread on a homemade selective solid medium, Mannitol Salt agar (MSA; Oxoid, Wade Road Basingstoke, UK) supplemented with 6% NaCl and cefoxitin (3.5 mg/L). Presumptive *Staphylococcus* colonies were identified phenotypically (cocci gram-positive, catalase-positive); five colonies from each plate, considered representative of the whole *Staphylococcus* population, were used for the subsequent analysis. The bacterial identification was confirmed by performing a phenotype analysis with a matrix-assisted laser desorption and ionisation time-of-flight mass spectrometry (MALDI-TOF MS) Microflex™ LRF (Bruker Daltonik GmbH, Bremen, Germany).

For genotypic analysis, the five original colonies were individually amplified onto a non-selective solid medium (Tryptic Soy Agar—TSA; Oxoid, Wade Road Basingstoke, UK) and incubated for 24 h at 37 ± 1 °C. DNA was extracted from bacterial colonies using an alkaline lysis-modified method described by Tramuta et al. [33]: briefly, one or two colonies obtained from amplification on the TSA medium were selected with a sterile loop and diluted in 50 µL of 0.5 M NaOH, and vortexed for 30 s. After 30 min, the same amount of tris(hydroxymethyl)aminomethane (Tris) buffer, 1 M, pH 7.5, was added and vortexed for 2 min. Finally, 50 µL of sterile water was added. DNA extracts were stored at −20 °C until further usage. A polymerase chain reaction (PCR) was implemented to verify the presence of the methicillin-resistance *mecA* gene (162 bp, amplicon size) [34]. Positive controls (*mecA*-positive *S. aureus* from Turin University Culture Collection) and negative controls (deionized DNA-free water) were added to every PCR reaction.

##### Extended-Spectrum β-Lactamase (ESBL) Producing *Escherichia coli*

For the recovery of ESBL-producing *E. coli* from environmental samples, we followed the protocol described by DTU Food-National Food Institute [35]. Briefly, 5 mL of buffered peptone water (BPW) was added to each sample for pre-enrichment. Samples were incubated at 37 ± 1 °C for 18 to 24 h. After 24 h, each specimen was plated using a loop of 10 µL onto US-1 MacConkey agar 3 (Oxoid, Wade Road Basingstoke, UK) supplemented with 1 mg/L cefotaxime and incubated at 37 °C for 18 to 24 h. Up to five morphologically consistent lactose-fermenting colonies, presumptive *E. coli* colonies were selected from each plate, and the identification was confirmed by MALDI-TOF MS, as described above for MRS. Phenotypic ESBL-*E. coli* confirmation was performed using the combination disc test method (Cefpodoxime Combination Disc Kit; Oxoid, Wade Road Basingstoke, UK) following the instructions. The reading was performed in accordance with EUCAST [36], which evaluates the inhibition of ESBL-*E. coli* activity by clavulanic acid. Briefly, Mueller-Hinton agar plates were inoculated with a 0.5 McFarland bacterial suspension to obtain a confluent growth, and the two antibiotic-containing discs were laid two centimetres apart. After 24 ± 1 h at 37 ± 1 °C, the inhibition halos were measured, and the isolate was defined as ESBL-producing when the difference between the halos of the cephalosporin alone and the corresponding one with clavulanate was greater than 5 mm.

### 2.3. Data Analysis

Each of the five zones of the farms obtained a biosecurity score, calculated through the sum of the scores attributed for each item by the BEAT application. The five zonal scores were then converted into a percentage (0–100), where a low percentage indicated a poor biosecurity status (i.e., insufficient measures) and a high percentage indicated a satisfying status (i.e., a full application of biosecurity measures) [12,37].

Statistical analysis was performed using the XLSTAT 2022.2.1 software (Addinson, TX, USA, 2022). As already applied by Scollo et al. [12], principal component analysis (PCA) and hierarchical cluster analysis (HCA) were used to describe the implementation of the five biosecurity zones shown by each farm during the 12 months of the study (active variables). Variation was expressed as the difference for each parameter between visit 3 and visit 1 (i.e., Δ = visit 3–visit 1). Variations of ATP values and total biosecurity scores (the average score of 5 zones) were included as supplementary variables. Factors with eigenvalues ≥ 1.0 (Kaiser criterion) were retained [38]. In the description of clusters, only variables with a square cosine > 0.2 were used [39]. A further descriptive analysis was performed using the Wilcoxon signed-rank test on data collected on the first and the third visit (i.e., pre- and post-intervention plan results) to identify changes in average scores and ATP values over time [40]. Differences between the productive phases (breeding/nursery vs. fattening) were performed by the Mann–Whitney test; differences among the ATP values of the 5 different positions in the pens were analysed by Kruskal–Wallis test for multiple comparisons, using the Dunn’s method and Bonferroni’s correction. Prevalence of positive MRS and ESBL-*E. coli* samples was calculated with a 95% confidence interval (95% CI). Farms and the different sampling sites were considered MRS- or ESBL-*E. coli*-positive when at least one of the (up to) five colonies processed per sample was positive. Fisher’s Exact test was used to assess differences in MRS and ESBL-*E. coli* prevalence before and after hygiene procedures and among different production stages.

## 3. Results

Among the 20 visited farms, 5 were breeding/nursery sites, and 15 were fattening sites. The average number of pigs reared in the fattening farms was 2406 ± 1978 standard deviation (minimum = 350; maximum = 6500), and 1840 ± 1071 (minimum = 800; maximum = 3600) in breeding/nursery sites. In the development of the 20 tailor-made plans, an average of 10.2 ± 6.8 recommendations were formulated per each farm. Among these, several recommendations were common for all the farms when not already applied (e.g., use only farm boots or disposable footwear), but others were customised depending on the farm context. For example, the hygiene lock for personnel and visitors was totally rebuilt in three farms which were available for economic investment, also providing sinks for hand hygiene and showers. Other 12 farms adapted the existing facilities to clearly separate dirty and clean areas (e.g., a bench to physically delimit the passage where footwear must be worn) in order to reduce expenses. In case of economic restrictions, none or only minor recommendations were adopted (five farms).

A descriptive analysis of the farms on the first and third visits is shown in Table 1. The average improvement at the end of the study regarding the total biosecurity score was 1.4 ± 0.9% (from 58.2 ± 9.6 to 59.6 ± 10.5%, *p* = 0.039; minimum −0.5%, maximum 6.7%), in which 13 farms improved their total biosecurity scores (65.0% of farms), 2 farms worsened their scores (10.0%), and 5 farms took no actions (25.0%). Regarding the five farm zones analysed, three showed a significant improvement in the biosecurity scores comparing the beginning and the end of the study: the public/professional transition improved by 1.1 ± 1.2%, the professional zone improved by 4.7 ± 2.2%, and the professional/herd transition improved by 1.4 ± 1.1% (Table 1). The practices with the greatest improvement belonging to these three zones were especially related to decreasing the persistence of pathogens in the barns (including the adoption of a rodent control plan implemented by a professional company) and reducing the risk of contamination by trucks through the hygiene procedures learned during the training (washing surfaces with high-pressure water and cleaning and disinfection of the loading bay; +11.9 ± 13.7%). The risk of staff contamination during the storage of dead animals also accounted for the improvement (cleaning and disinfection of surfaces and equipment after carcass management; +6.2 ± 13.9%). This improvement was reflected by the ATP values detected in the environmental swabs. Indeed, Wilcoxon signed-rank test identified significant changes of average values over time in this parameter that decreased from 2539.7 ± 2025.2 to 1696.1 ± 1160.1 RLUs (*p*-value = 0.039; reduction observed in 19/20 farms). Among the farms that decreased their total biosecurity scores, the practices that worsened more were related to the procedures of carcass elimination from the farm (−7.9 ± 11.4%) and procedures of the entrance of personnel and visitors (−3.0 ± 7.3%)

Breeding/nursery farms showed a higher mean biosecurity score than fattening farms (66.6 ± 9.2 vs. 56.9 ± 8.8%; *p*-value = 0.009) but also a higher ATP level (3339.3 ± 2112.4 vs. 1605.2 ± 1178.5 RLUs; *p*-value = 0.014). Farm zones that showed a significant statistical difference between productive phases were the public (*p*-value = 0.037), the professional/herd transition (*p*-value < 0.0001), and the herd zone (*p*-value = 0.012). Boxplots of the distribution of biosecurity scores for farm zones in breeding/nursery and fattening sites are shown in Appendix A.

The PCA and HCA (Figure 2) carried out on the variation of both active (biosecurity scores of each zone) and supplementary variables (total biosecurity scores and ATP levels) identified two clusters of farms (A, B). Changes in the biosecurity scores and in the ATP levels defining the clusters are shown in Table 2, and a brief description is reported in the text below. From the cluster analysis, the first two factors (eigenvalues ≥ 1.0) synthetised by PCA accounted for 75.9% of the variability. All the variables were considered in the description as all of them showed a squared cosine > 0.2, suggesting a relevant contribution to the two factors.

Cluster A: farms that generally improved their total biosecurity scores. Six farms belonged to this cluster, and they excelled in improving their biosecurity score, concentrating efforts on the professional zone (+12.0%). A pronounced reduction in ATP levels from environmental swabs was observed.

Cluster B: farms with a limited improvement of their total biosecurity scores. Most of the farms involved in the study belonged to this cluster (14 farms, 70.0%), showing a scarce improvement in the total biosecurity score after the 12-month period of the study. Even if they succeeded in marginally improving their professional zone score, this was associated with worsened scores in the public and the herd zones. A slight reduction in ATP levels was observed.

ATP levels from environmental swabs in the five different positions of the pens showed a significant difference (*p*-value = 0.048), with the feeding tube being at the highest levels (3152.3 ± 2997.2 RLUs) (Figure 3).

During the first visit in the subsample of 10 farms (4 breeding/nursery and 6 fattening sites), 27 MRS were recovered from 40 environmental swabs before hygiene procedures, right after emptying of the pen from animals (67.5%; 95% CI: 50.9–81.4), as well as 12 ESBL-*E. coli* (30.0%; 95% CI: 16.6–46.5; Appendix A). Among the 27 positive MRS samples, 9 were positive for LA-MRSA (22.5%; 95% CI: 10.8–38.4). After the complete hygiene procedures, still on the empty pen, MRS and LA-MRSA prevalence did not decrease (MRS, 26 positive samples over 40 = 65.0%; 95% CI: 48.3–79.4; LA-MRSA, 4 positive samples over 40 = 10.0%; 95% CI: 2.8–23.7; *p*-value > 0.05), whereas ESBL-*E. coli* prevalence dropped dramatically (1 positive samples over 40 = 2.5%; 95% CI: 0.06–13.2; *p*-value = 0.001). A significant decrease in ESBL-*E. coli* prevalence was also shown in the corner of the dunging area (*p*-value = 0.018). Both LA-MRSA prevalence and ESBL-*E. coli* prevalence for each sampling site collected during the first visit are presented in Figure 4. In the breeding/nursery sites, the prevalence before hygiene procedures of MRS at the first visit was 75.0% (95% CI: 47.6–92.7), while it was 81.2% (95% CI: 54.3–95.9) after cleaning and disinfection (*p*-value > 0.05); LA-MRSA prevalence was 12.5% (95% CI: 1.5–38.3) and 6.2% (95% CI: 0.2–30.2), respectively (*p*-value > 0.05); regarding ESBL-*E. coli*, the prevalence rates were 37.5% (95% CI: 15.2–64.6) and 0.0% (95% CI: 0.0–0.2), respectively (*p*-value > 0.05). In the fattening sites, the prevalence before hygiene procedures of MRS at the first visit was 62.5% (95% CI: 40.6–81.2), and it became 54.2% (95% CI: 32.8–74.4) after cleaning and disinfection (*p*-value > 0.05); LA-MRSA prevalence was 29.2% (95% CI: 12.6–51.1) and 12.5% (95% CI: 2.7–32.4), respectively (*p*-value > 0.05); regarding ESBL-*E. coli*, as above, the prevalence rates were 25.0% (95% CI: 9.8–46.7) and 4.2% (95% CI: 0.1–21.1) (*p*-value > 0.05). Prevalence rates of MRS and ESBL-*E. coli* for each farm (*n* = 10) collected at the beginning of the study (first visit) are reported in Appendix A. Isolations from environmental swabs collected at the end of the study (third visit) were also reported for six farms. The prevalence of MRS before and after hygiene procedures at the third visit was not statistically significant (*p*-value > 0.05). In total, 189 MRS colonies from 80 swab samples were investigated: *Staphylococcus aureus* was isolated 45 times (22.6%). *Staphylococcus sciuri* (29.6%), *S. saprophyticus* (20.6%), *S. equorum* (12.1%), *S. haemolyticus* (4.5%), *S. xylosus* (3.5%), and *S. epidermidis* (2.5%) were also isolated.

## 4. Discussion

This study describes the strengths and weaknesses of biosecurity in pig farms through the application of the BEAT, a biosecurity checklist specific for swine facilities based on the FAO three-zone biosecurity model [12]. The tool was used to develop tailor-made plans to implement biosecurity in a convenience sample of 20 farms during a 12-month period, with a particular focus on hygiene management as an important biosecurity vehicle. Personnel training on hygiene procedures with practical sessions on-farm, under the guidance of an expert technician, was part of the tailor-made plan. The study resulted in two farm profiles based on biosecurity improvement over 12 months and, for each farm profile, a description of the ATP content in the environment after hygiene procedures was provided and used as a biomarker output. To gain an insight into the environmental samples, the presence of MRS (with a specific focus on LA-MRSA) and ESBL-*E. coli* was investigated in a subsample of 10 farms, as LA-MRSA and ESBL-*E. coli* are considered sentinel microorganisms to monitor antibiotic resistance in farm animals as well as in humans.

Even if a general positive progress of the total biosecurity score was observed after the 12-month period, the professional zone (i.e., between the animal barns) and its transition to/from the public and herd zones were the areas that achieved the major improvement during the study. In general, implementation driven by a tailor-made plan is more likely to be reached compared to a standardised approach [12,25], as a customised approach increases the likelihood of realistic implementation of the biosecurity layout. In fact, only the involvement in the planning phase of personnel that will become “actors in the evolution” can fill the biosecurity gaps of the farm and identify “best practices” for improvement. The specific upgrading observed in the professional zone and its transition to/from the public and herd zones in the present study might be related to the historical lack in farms of a proper and organised farm design and planimetry, specifically planned to control disease or to increase management efficiency, as already observed by Da Costa et al. [41]. In fact, the expansion of many farms during the last few years by adding new buildings to older ones led to the impossibility (and/or the lack of interest) to valorise the pertinence of internal biosecurity in the past. Actually, the public zone and their transitions showed the lowest score at the beginning of the study.

Among the measures applied to these improved zones, the practices with the greatest improvement were those related to decreasing the persistence of pathogens (+11.9%). The improvement of hygiene management was the first practice aimed at decreasing the persistence of pathogens, supporting the importance of personnel training (lectures and practical assistance on-farm) that was included in the tailor-made biosecurity plans of the present study. Over the past few years, some studies explored the factors that affect the decision making of swine farmers and their attitude towards biosecurity [11,42]. All the authors agreed that biosecurity practices and hygiene procedures are often adopted by pig farmers with low perseverance and consistency, although their importance in preventing and controlling diseases is acknowledged. Failure to comply with biosecurity measures seems to be frequently associated with inadequate training of farm personnel and their poor communication with the advisors [11,43], in particular, as regards the understanding of the significance of each measure in terms of disease transmission. In the present study, it was supposed that the training of the personnel and the application of a tailor-made biosecurity plan increased the communication between farmers and biosecurity advisors, in particular on those measures most related to the training scope (i.e., hygiene procedures). Probably, planning three visits over the 12-month period was also helpful in leading farmers to pay more attention to recommendations from the advisor. Moreover, the use of a simple and rapid biomarker as the ATP content to evaluate the amount of organic matrix in the environment after hygiene procedures was useful to objectively quantify the results of the cleaning process and increase the awareness of the farmer directly on-farm. Targeted training in combination with monitored results can help to increase efficiency and prevent it from becoming inattentive due to routine. This suggestion is in agreement with clusters of farms obtained in the present study, which distinguished farms that greatly improved their total biosecurity scores (and showed a pronounced reduction in ATP levels from environmental swabs over the 12-month period) and farms with a scarce improvement of the total biosecurity scores (and a scarce reduction in ATP levels as well). The training effect regarding hygiene procedures was already described by Heinemann et al. [31], who observed a decrease in environmental ATP levels in pig facilities. Actually, the authors suggested the development of a tailor-made hygiene protocol to be applied on-farm in consultation with a supervising veterinarian as a possible route for improving hygiene management; this protocol would be personally self-monitored by the personnel carrying out the work, similar to a hazard analysis critical control point (HACCP) system already existing in the food industry. A continuous training on-farm, perhaps guided by an expert with a possible turnaround of once a year, represents a plausible opportunity for improvement. Other authors suggested that the key to success was to make farmers aware that the use of detergents to clean the barns better prepares surfaces for subsequent disinfection [44]; a successful application of cleaning and disinfection procedures always depends on the high competence and improved skills of the person performing the work [45,46,47]; time is a key factor that affects the efficiency of hygiene procedures, and that the knowledge of farm-specific weak points in cleaning and disinfection procedures is fundamental [48]. Moreover, to improve hygienic management on farms, it is important to change the attitude of the farmers using persuasive arguments [31,49], such as economic aspects led by greater productivity but also the improvement of animal welfare [47,50,51]. Precisely in this regard, some authors have recently strengthened the positive effects of tailor-made plans on biosecurity in pig farms by using antimicrobial use and economic and technical performances as output parameters. Following interventions, a substantial reduction in antimicrobial use was achieved, maintaining the overall farm technical performance, and a possible link between biosecurity and lung lesions and scars at slaughter has been proposed [9,10,11,12].

Another practice that was part of the measures to decrease the persistence of pathogens was the rodent control plan, which was implemented together with the reduction of staff contamination practices during the storage of dead animals (+6.2%). Both these measures were also implemented in the previous study of Scollo et al. [12], which highlighted the importance of rodents as a vector for numerous pathogens that affect pigs, such as *Campylobacter* spp., *Yersinia pseudotuberculosis*, some *Salmonella* serovars, the encephalomyocarditis virus, *Leptospira* spp., *Toxoplasma gondii*, *Brachyspira* spp. or *Lawsonia intracellularis* [52]. Moreover, to decrease the risk of staff contamination during carcass management, the positioning of the carcass storage is encouraged outside of the external perimeter of the farm to avoid the need to enter the farm. Truck drivers and farm personnel involved in carcass management should never be allowed to cross the clean areas of the herd and the professional zones, as small amounts of organic matrix on their equipment could be sufficient to colonise a farm [52]. However, worsened carcass management was the cause of the biosecurity score reduction in one of the two farms that showed a worsened score on the third visit. In fact, technical issues related to the well-functioning of carcass storage increased the frequency of carcass elimination by trucks. Moreover, a second issue in the 12-month period that led to a worsened score was the reduction of the proper application of the right procedures for personnel and visitors’ entrance. The cause was the turnover of personnel and the hiring of less experienced employees. The presence of a cluster (B) with 14 farms that scarcely improved their total biosecurity score (including two farms that worsened) might reflect one of the main problems for such long-term programs: a positive result is the absence of new diseases that enter the farm or the absence of internal spread of pre-existing ones. Differently, in the case of an ineffective program, the likelihood of a new disease is unpredictable and might occur after an undetermined time frame or not occur at all. This means that if the program is successful, nothing will take place, as well as in case of an unsuccessful program until an unpredictable outbreak. The concept might be totally confounding for farm personnel, leading to a low perception of the risk, the slowdown in biosecurity implementation, and the perception of the uselessness of the measures [4].

The results showed that breeding/nursery farms totalized a higher mean biosecurity score than fattening farms, according to Silva et al. [53] and Scollo et al. [54], since these were most likely to undergo certification and annual monitoring by the official veterinary service. Moreover, breeding farms are at the top of the sanitary pyramid in pig production; they have a high sanitary status and a reduced risk of the introduction of pathogens [55]. These observations suggested that stringent monitoring conducted for breeding/nursery is likely more effective as opposed to less monitored fattening farms. However, ATP levels in these farms were the highest. Indeed, several structures that furnish the farrowing rooms and the nursery facilities (e.g., crates, bowls, sloped walls, rails, and raised bars) increase the complexity of hygiene procedures and dirtiness removal.

Findings related to nipple drinkers and feeding tubes suggested that these points must be considered the most critical in the hygiene process, as also reported by other authors [31,56]. Mannion et al. [57] measured the *Enterobacteriaceae* counts in pig finisher farms and showed that feeders and drinkers were more contaminated after cleaning and disinfection procedures than floors, which is comparable to these results. They suggested that feeders and drinkers are re-soiled during washing with high-pressure water due to the squirting of contaminated liquids. Probably, as both the drinkers and the feeding tubes contain internal surfaces not exposed to standard cleaning procedures, a specific hygiene plan for water and feeding lines should be improved, also to decrease the risk of biofilm development [58]. Gonzalez et al. [59] demonstrated that because of the improper cleaning of drinkers, the bacterial load detected in the analysed drinking water samples for livestock use was high. The contaminated drinkers and feeders might easily transfer pathogenic bacteria from animals of the previous batch to newly arriving pigs via water or feed intake. For example, the importance of effective hygiene procedures to avoid the carryover of *Salmonella* in livestock has been demonstrated by several authors [45,46,60]. This consequence can also be applied to LA-MRSA and ESBL-*E. coli*, whose importance is related both to animal and human health, as they are considered indicator organisms for resistant bacteria [61]. In livestock production, the occurrence of MRS and ESBL-*E. coli* depends, in addition to antibiotic usage, on the amount of dust and faeces [62,63,64]. MRS and ESBL-*E. coli* were also found in the farms involved in the present study. MRS prevalence in dirty pens (67.5%) was in line with the results of Bonvegna et al. [32], who reported a prevalence of 64.6% in intensive and organic pig farms in Italy but was higher than those reported in previous studies from other European countries, where a prevalence ranged from 36.3% to 6.5% was reported (in Switzerland and Belgium, respectively [65,66]). Regarding the specific case of LA-MRSA, the prevalence observed in the present study (22.5%) was strongly higher than the prevalence reported by Bonvegna et al. [32], who reported no positive environmental samples and only one positive sample from nasal swabs over 195 tested animals. However, only five farms were involved, including an organic farm and an antibiotic-free growing site, probably less predisposed to the occurrence of LA-MRSA. Our observed prevalence was also higher than those reported in pig fattening farms in another study conducted in the north of Italy [67], which were estimated at 6.7% in the environment and 17.5% among animal samples. Actually, an even higher prevalence was reported in the south of Italy, where more than 45% was reported in animals, in particular in intensive production systems [68,69]. ESBL-*E. coli* prevalence (30.0%) was in agreement with some results of other authors, who reported a prevalence ranging from 3.2 to 35.0% [17,70] but was higher than those reported by the EFSA report [71], which showed an average prevalence of 2.3%, ranging from 0 to 3.8% in different European countries. Unlike ESBL-*E. coli,* whose prevalence was effectively decreased by extensive hygiene procedures (in agreement with the results of Schmithausen et al. [17]), a challenging issue regarding the persistence of a still high prevalence of LA-MRSA after cleaning emerged from the present study. In fact, both at the beginning of the study and 12 months later, hygiene procedures were not able to decrease its prevalence in the environment, despite the evident decrease of the ATP content in the environment. Similar results were reported by other authors on different livestock species [72,73]. Several hypotheses should be considered for this result. For example, Schmithausen et al. [17] successfully decontaminated pig farms from LA-MRSA, including a complete depopulation in their protocol. As LA-MRSA can be transmitted via air and dust and is emitted via ventilation systems into the ambient air [63,74,75], it is not possible to exclude that a re-contamination occurred from the nearby populated barns during the drying time after the hygiene process. This might explain why some farms, at the beginning of the study, showed a higher prevalence of positive MRS samples after hygiene procedures than before (Appendix A, farms n. 3, 7, 8, 10). Kobusch et al. [76] suggested that farms resulting positive for LA-MRSA after an extensive hygiene process might be re-contaminated by boots or clothing and cross-contamination caused by personnel. Moreover, an inappropriate selection of a suitable disinfectant might be responsible for the failure, as all the farmers were trained to use a disinfectant during the study, but no specific component was mandatory in the protocol. For example, quaternary ammonium compounds (QAC) are usually used for disinfection in animal husbandry. The possible role of QAC-resistance genes, already described in hospital isolates of *S. aureus* as usually related to plasmids carrying antimicrobial-resistance genes [77,78], should be investigated. Lastly, LA-MRSA is a biofilm-producing pathogen and has become notorious for its persistence and for causing chronic infections in hospitals due to its ability to resist therapeutic treatment by forming biofilms on indwelling medical devices, including implanted artificial heart valves, catheters, and joint prosthetics [79]. Besides several hypotheses for its persistence, the presence of LA-MRSA surely poses a hazard for people working in swine facilities and for citizens living in areas with a high density of pig farms [80,81,82].

Among the MRS isolated in the present study, *S. sciuri* was the species more frequently detected (29.6%). Consistently with previous research on animals, sewage, and dust in pig farms in Asia and Europe [83,84], Bonvegna et al. [32] put forth a hypothesis suggesting that the prevalence of *S. sciuri* in the farm environment is related to the contamination of surfaces during the rooting behaviour of the pigs, whose upper respiratory tract is usually colonised by the pathogen. Notably, *S. sciuri* is a versatile microorganism that is well-adapted to a variety of hosts and can thrive as a free-living organism [85]. The relevance of *S. sciuri* is related to its ability to carry a wide variety of antibiotic resistance and virulence genes and to transfer them to other members of the Staphylococci group, including LA-MRSA, by horizontal gene transfer. It has been isolated from a variety of sources such as soil, water, faeces, and foods, strengthening the hypothesis that *S. sciuri* is a reservoir of these antibiotic-resistance genes; a systematic inclusion of this pathogen in monitoring plans warrants a more comprehensive approach to antibiotic resistance in food animal production [86]. Another MRS frequently isolated in the present study was *S. saprophyticus* (20.6%). It has been recognised as an important opportunistic pathogen, and its relevance is mainly related to its role in human health, as it is one of the pathogens more frequently recognised as the causal agent of infections of the urinary tract [87,88].

Results regarding the high prevalence of LA-MRSA and ESBL-*E. coli* in the sample of farms, and the failure in decreasing the presence of LA-MRSA, and MRS in general, in the barn environment, arose the challenging condition about the spread of livestock-associated resistant pathogens to humans, which is well-known in the current era [80,81,82]. The major challenge is the silent spread of colonising multidrug-resistant pathogens among human hospitalised patients with evident risk for acquisition of resistant bacteria and—even worse—into those with no history of hospitalisation or travel. MRS strains have been found throughout all different levels of the pig production chain. Remarkably, strains of LA-MRSA have been detected in people who work in close contact with pigs, and they are more frequently identified in hospitals situated in rural areas, leading to the issue of the healthcare of workers [17].

A limitation of the study was the unavailability of antimicrobial consumption in the recruited farms and as high prevalence of MRS and ESBL-*E. coli* might be driven by high antimicrobial usage. Moreover, the possible involvement of the use of heavy metals with antimicrobial activity as feed supplements should not be excluded in the selection of MRS [89], as the data collection in farms was performed before the legal ban on the medicinal use of Zinc oxide from farming pigs [90] (effective from June 2022).

## 5. Conclusions

The usefulness of a systematic evaluation of biosecurity seems to be enhanced when a tailor-made biosecurity plan is formulated, monitoring its implementation. This approach, together with a training on-farm regarding hygiene procedures, is suggested to lead to a general improvement in biosecurity, with particular effect on the professional zone and their transitions as the main recipient of cleaning and disinfection procedures. To better drive farmers towards consciousness through objective outputs, ATP rapid test in the environment after hygiene procedures might be a useful biomarker, as it can easily check the efficacy of the procedures on the farm. However, the successful improvement of the biosecurity score in specific zones and the reduction of the ATP content after the 12 months of the study should not reduce the attention to the presence of LA-MRSA and ESBL-*E. coli* in the pig barns. In fact, the hygiene procedures applied were able to reduce the prevalence of ESBL-*E. coli* but not that of LA-MRSA, neither at the beginning of the study nor after the application of the tailor-made plans. More efforts are needed to manage this threat through specific protocols, in particular, because of the risk of spreading this resistant pathogen to humans in close contact with pigs, moving the attention to the healthcare of workers in a one-health approach.

## Figures and Tables

**Figure 1 animals-13-01262-f001:**
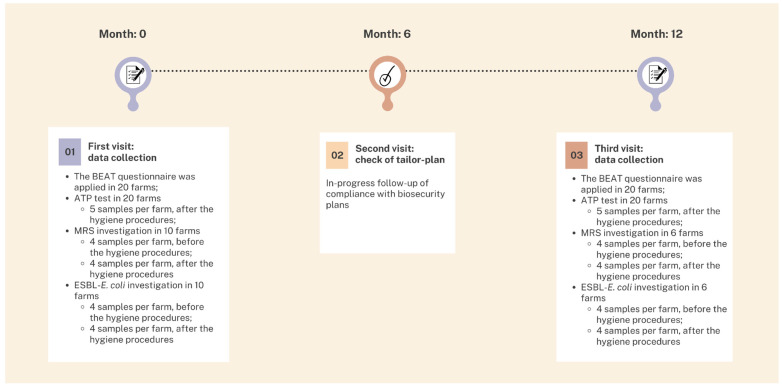
Timeline of the data collection over the 12-month period of the study. BEAT: Biosecurity Risk Analysis Tool; ATP: adenosine triphosphate; MRS: methicillin-resistant staphylococci; ESBL-*E. coli*: extended-spectrum β-lactamase producing *Escherichia coli*.

**Figure 2 animals-13-01262-f002:**
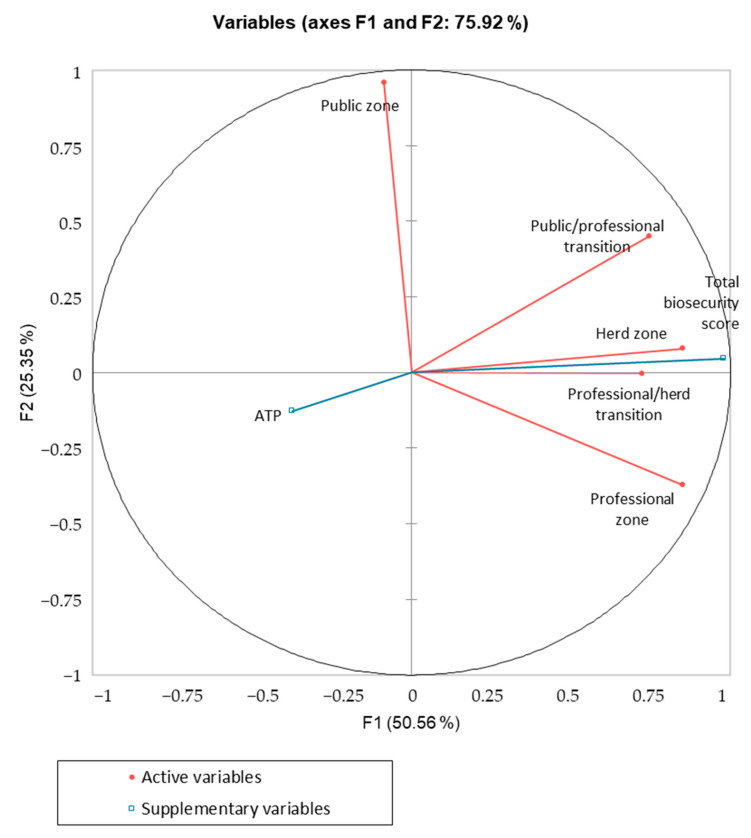
Correlation circle of PCA analysis. Active variables are changes in biosecurity scores in each zone after 12 months and are represented in red: public zone, public/professional transition, professional zone, professional/herd transition, and herd zone. Supplementary variables are represented in blue.

**Figure 3 animals-13-01262-f003:**
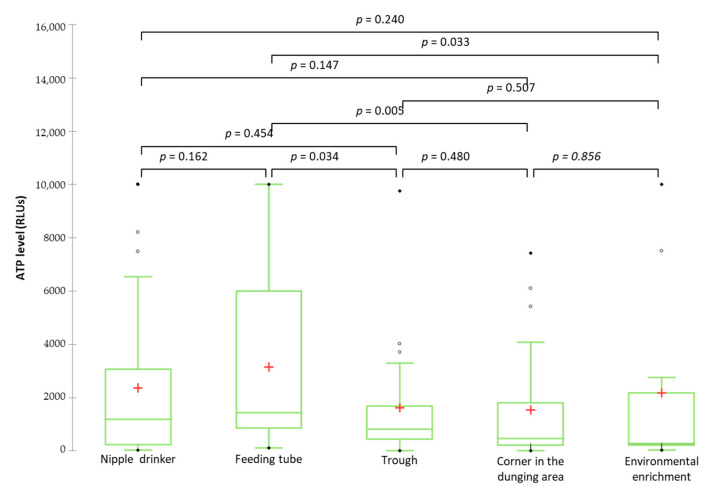
Boxplots of the distribution of ATP levels from environmental swabs for each sampling site in 20 pig farms. In the boxes, the thick horizontal line represents the median biosecurity score, whereas the base and the top of the boxes are the first and third quartiles, respectively. Whiskers extend to a maximum of 1.5 times the interquartile range. Red crosses represent the mean, black dots represent extreme values (maximum and minimum), and hollow circles are the outliers.

**Figure 4 animals-13-01262-f004:**
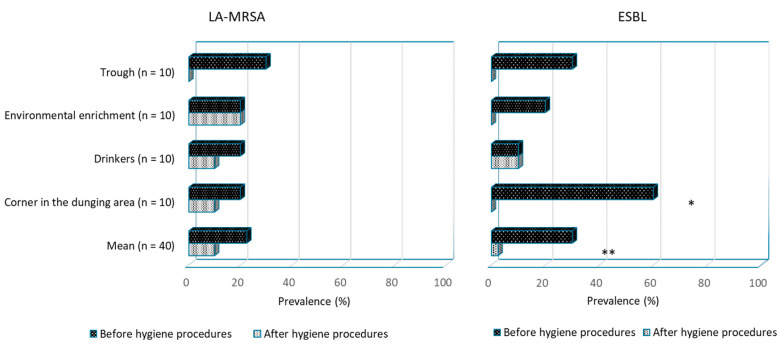
Prevalence at the beginning of the study (first visit) of livestock-associated methicillin-resistant *Staphylococcus aureus* (LA-MRSA) and extended-spectrum β-lactamase (ESBL) producing *E. coli* in environmental samples for each sampling site in 10 pig farms. * *p*-value < 0.05; ** *p*-value < 0.01.

**Table 1 animals-13-01262-t001:** Descriptive analysis of the farms (*n* = 20) at the beginning and at the end of the study: biosecurity scores for each of the 5 zones and adenosine triphosphate (ATP) values. RLUs = relative light units. Ns = Not statistically significant.

	First Visit Mean ± sd (Minimum–Maximum)	Third Visit Mean ± sd (Minimum–Maximum)	*p*-Value
Biosecurity scores			
Public zone (%)	63.9 ± 12.7 (43.8–87.5)	63.5 ± 12.1 (47.9–87.5)	ns
Public/professional transition (%)	57.7 ± 16.9 (25.0–86.5)	58.8 ± 18.1 (25.0–90.6)	0.040
Professional zone (%)	57.4 ± 13.3 (35.0–85.0)	62.1 ± 15.5 (37.5–92.5)	0.003
Professional/herd transition (%)	39.9 ± 8.3 (27.6–52.6)	41.3 ± 9.4 (27.6–56.6)	0.027
Herd zone (%)	71.9 ± 8.5 (58.3–90.7)	72.2 ± 9.0 (58.3–92.6)	ns
Total biosecurity score (%)	58.2 ± 9.6 (38.7–79.7)	59.6 ± 10.5 (39.5–84.0)	0.006
ATP (RLUs)	2539.7 ± 2025.2 (143.2–8167.5)	1696.1 ± 1160.1 (265.2–3739.7)	0.039

**Table 2 animals-13-01262-t002:** Changes in biosecurity scores after 12 months and ATP level evolution in the two clusters identified with the HCA. All the variables showed a squared cosine > 0.2. RLUs = relative-light units.

Item	Cluster A	Cluster B
N. farms	6	14
Biosecurity score changes		
Public zone (%)	0.0 ± 0.0	−0.4 ± 3.8
Public/professional transition (%)	3.3 ± 0.9	0.3 ± 1.9
Professional zone (%)	12.0 ± 6.9	2.3 ± 3.8
Professional/herd transition (%)	4.5 ± 3.9	0.3 ± 1.0
Herd zone (%)	1.8 ± 1.1	−0.1 ± 0.8
Total biosecurity score (%)	4.3 ± 1.5	0.4 ± 1.0
ATP (RLUs)	−2837.9 ± 1978.4	−700.0 ± 1860.6

## Data Availability

Not applicable.

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
