# Peer review of "Biosecurity and Hygiene Procedures in Pig Farms: Effects of a Tailor-Made Approach as Monitored by Environmental Samples"

_animals, 2023, doi:10.3390/ani13071262_

Round 1
Reviewer 1 Report
The manuscript gives some interesting new sights in biosecurity and hygiene procedures on pig farms.
Before I can recommend the manuscript for publication, some additions and clarifications of content should be made. Some general comments are given below. Other remarks are commented directly in the uploaded pdf file.
- It is not clear from the manuscript what the tailor-made adaptation to the particular operation consists of.
For the direct comparison it is unfavorable that the farm types are not equally distributed (instead 5 breeders and 15 fatteners). It is also not clear whether the questions for the biosecurity score are the same for breeders and fatteners.
How do you estimate the reduction of the biosecurity score? The decrease is significant, but does it manifest in improved animal health? Is the reduction a substantial improvement and worth the effort of the tailored plan?
How do you estimate the effect of having the third (last) farm visit announced? Could this have an effect on the results? This should be discussed as well.

Author Response
We thank the reviewer for her/his careful comments. In agreement with the suggestions of all the reviewers and the Editor, we have profoundly edited the text and answered to the clarifications (see attached file).
- The entire document is now focused on livestock-associated methicillin-resistant Staphylococcus aureus (LA-MRSA) and extended spectrum β-lactamase producing Escherichia coli (ESBL- coli) as sentinel microorganisms to monitor antibiotic resistance. The other MRS have been discussed as well but as a secondary result. However, their relevance has been added in the text;
- An improved description of the tailor-made plan has been added, including same examples. In addition, the training procedures has been detailed. Emphasis was given to the aim of monitoring the improvement of the pre-existing hygiene procedures, suggesting and training the personnel on the correct one that was the same for all the farms;
- Conclusion has been modified, and more emphasis was given to the improvement of professional zone (and their transitions) as a result of the study, instead of the general biosecurity improvement.

Reviewer 2 Report
The manuscript "Biosecurity and hygiene procedures in pig farms: effects of a 2 tailor-made approach as monitored by environmental samples" by Scollo et al describes the use of BEAT based custom designed methods to improve biosecurity of the swine farms. A major concern is that thre conclusion are not fully supported by the results obtained and must be thoroughly revised. In addition, sanitation and disinfection protocols/methods for each must be described in order to correctly asses the impact on overall biosecurity and to understand develop models for better management systems.
Addtional comments are in the attached annotated pdf file.

Author Response

(The authors gave the same response as above.)

Reviewer 3 Report
This is an important study in the world of livestock biosecurity testing tailor-made plans to improve biosecurity in pig farms which eventually impacts workers health. I have included some general comments as well as specific comments to consider.
General comments:
Why was Salmonella not considered as one of the microorganisms to monitor antibiotic resistance?
Specific comments:
Line 10: ATP appearing for the first time hence spell it out and put (ATP) in the parenthesis.
Line 282: Kruskal-Wallis test has to be corrected for spell check.
Line 291, 292: Are those values mean+/-SD? If so, please mention
Figure 1 and 3: Make the boxplots clear
Author Response
We thank the reviewer for her/his careful comments. In agreement with the suggestions of all the reviewers and the Editor, we have profoundly edited the text and answered to the clarifications (see the attached file).
- The entire document is now focused on livestock-associated methicillin-resistant Staphylococcus aureus (LA-MRSA) and extended spectrum β-lactamase producing Escherichia coli (ESBL- coli) as sentinel microorganisms to monitor antibiotic resistance. The other MRS have been discussed as well but as a secondary result. However, their relevance has been added in the text;
- An improved description of the tailor-made plan has been added, including same examples. In addition, the training procedures has been detailed. Emphasis was given to the aim of monitoring the improvement of the pre-existing hygiene procedures, suggesting and training the personnel on the correct one that was the same for all the farms;
- Conclusion has been modified, and more emphasis was given to the improvement of professional zone (and their transitions) as a result of the study, instead of the general biosecurity improvement.

Round 2
Reviewer 1 Report
The authors have implemented the comments very well and carefully revised the whole manuscript. The article is now very interesting and improves the level of knowledge about the implementation of hygiene measures in pig farming. Therefore, I have only small comments for improvement:
In some part the use of MRS and MRSA is still pretty mixed up. In line 28 you write that MRSA was investigated as sentinel organism. Does the result in Line 33 now refer to MRS or MRSA? This inconsistency is found in the simple summary, in the abstract and in some places throughout the manuscript (besides the result section, which is appropriately rewritten) and again requires revision. Perhaps the relevance of detecting MRS in general, would be better understood, if the authors add a sentence in the literature section about the role of staphylococci (other than S. aureus) in the spread of antibiotic resistance in agriculture. This aspect is explained very clearly in the discussion, but it would be helpful to mention this beforehand.
Table 2, 3rd row: Change "Biosecurity scores changes" to "Biosecurity score changes"
L479: delete the surplus 'the'
Author Response
We would like to thank again the reviewer for the suggestions. All the sentences have been addressed:
In some part the use of MRS and MRSA is still pretty mixed up. In line 28 you write that MRSA was investigated as sentinel organism. Does the result in Line 33 now refer to MRS or MRSA? Sorry, it refers to MRSA like the sentence before. Corrected. This inconsistency is found in the simple summary, in the abstract and in some places throughout the manuscript (besides the result section, which is appropriately rewritten) and again requires revision. Revised. Sorry, it was a refuse. Perhaps the relevance of detecting MRS in general, would be better understood, if the authors add a sentence in the literature section about the role of staphylococci (other than S. aureus) in the spread of antibiotic resistance in agriculture. This aspect is explained very clearly in the discussion, but it would be helpful to mention this beforehand. Sentence added L96-98.
Table 2, 3rd row: Change "Biosecurity scores changes" to "Biosecurity score changes" Done
L479: delete the surplus 'the' Done
Reviewer 2 Report
Revised manuscript has addressed the comments.
Author Response
We would like to thank the reviewer for the efforts in improving the paper.